# Increased Low Degree Spherical Harmonic Influences on Polar Ice Sheet Mass Change Derived from GRACE Mission

**Xiaoli Su** [1,2,*] **, Junyi Guo** [3]**, C. K. Shum** [3,4] **, Zhicai Luo** [1,2] **and Yu Zhang** [3]

1. The MOE Key Laboratory of Fundamental Physical Quantities Measurement and Hubei Key Laboratory of Gravitation and Quantum Physics, PGMF and School of Physics, Huazhong University of Science and Technology, Wuhan 430074, China; zcluo@hust.edu.cn
2. Institute of Geophysics, Huazhong University of Science and Technology, Wuhan 430074, China
3. Division of Geodetic Science, School of Earth Sciences, The Ohio State University, Columbus, OH 43210-1398, USA; guo.81@osu.edu (J.G.); ck.shum@outlook.com (C.K.S.); zhang.6345@osu.edu (Y.Z.)
4. Institute of Geodesy and Geophysics, Chinese Academy of Sciences, Wuhan 430077, China
* Correspondence: xlsu@hust.edu.cn

**Abstract:** Replacing estimates of $C_{20}$ from the Gravity Recovery and Climate Experiment (GRACE) monthly gravity field solutions by those from satellite laser ranging (SLR) data and including degree one terms has become a standard procedure for proper science applications in the satellite gravimetry community. Here, we assess the impact of degree one terms, SLR-based $C_{20}$ and $C_{30}$ estimates on GRACE-derived polar ice sheet mass variations. We report that degree one terms recommended for GRACE Release 06 (RL06) data have an impact of 2.5 times more than those for GRACE RL05 data on the mass trend estimates over the Greenland and the Antarctic ice sheets. The latest recommended $C_{20}$ solutions in GRACE Technical Note 14 (TN14) affect the mass trend estimates of ice sheets in absolute value by more than 50%, as compared to those in TN11 and TN07. The SLR-based $C_{30}$ replacement has some impact on the Antarctic ice sheet mass variations, mainly depending on the length of the study period. This study emphasizes that reliable solutions of low degree spherical harmonics are crucial for accurately deriving ice sheet mass balance from satellite gravimetry.

**Keywords:** GRACE; polar ice sheets; mass change; low degree spherical harmonic coefficients

## 1. Introduction

With its launch in March 2002, the Gravity Recovery and Climate Experiment (GRACE) mission provided unique observations about the Earth's temporal gravity field with unprecedented accuracy [1]. Its data record covered more than 15 years, largely improving our understanding of terrestrial water storage change, ice sheet mass balance, and glacier mass variations, as well as sea level change [2]. GRACE-based temporal gravity field solutions in the standard Level 2 (L2) data product consist of a series of spherical harmonic coefficients with degree and order up to at least 60. Due in part to the geometry of satellite constellation, it was stated early in the mission that the GRACE-determined degree-2 zonal term, $C_{20}$, was not reliable [3,4]. Previous studies reported that an unexpected ~161-day periodic signal was contained in GRACE-determined $C_{20}$ values, with the cause attributed to aliasing of the S2 ocean tide [5,6]. It is worth noting that this is not a consensus view. The authors of [7] suggested a cause associated with the temperature-dependent systematic error in the accelerometer data. Regardless of the true cause, it has become a standard procedure to replace GRACE-determined $C_{20}$ values with those provided by satellite laser ranging (SLR) data in order to accurately derive

mass redistribution within the Earth's system using GRACE monthly gravity field solutions. Indeed, SLR observations to geodetic satellites include valuable information about the low degree/order components of the Earth gravity field [8,9]. SLR can accurately measure the long-term variations in $C_{20}$, which provides an independent constraint on GRACE-determined $C_{20}$ [10]. Previously, the estimates of $C_{20}$ from SLR were documented in GRACE Technical Note (TN), together with a version number. For instance, TN07 for GRACE L2 Release 05 (RL05) data [11] and TN11 for GRACE Release 06 (RL06) data [7]. However, as the authors of [7] mentioned, there was a concern that SLR-provided $C_{20}$ values could be inconsistent with the other spherical harmonic coefficients in GRACE monthly solutions. They compared $C_{20}$ estimates from SLR only and from the combination of SLR and GRACE, with the results concluding that estimates of $C_{20}$ from SLR only were appropriate for science applications of GRACE data. This did not dispel the concern that the SLR $C_{20}$ estimates could not be used to stand in for GRACE [12]. The authors of [13] recommended a new $C_{20}$ product which was obtained by including GRACE-derived time-variable gravity in the SLR data reduction forward modeling. This new $C_{20}$ product was produced at the Goddard Space Flight Center (GSFC) and documented in TN14 for GRACE RL06 and GRACE-Follow-On (GRACE-FO) data. Besides, estimates of SLR-based $C_{30}$ (GSFC $C_{30}$) since March 2012 are also provided in TN14, aiming at improving the accuracy of GRACE-derived mass redistribution during the degradation of GRACE $C_{30}$ for single accelerometer mode [14]. However, some recent studies adopted the GSFC $C_{30}$ product in TN14 since March 2012 instead of GRACE-determined $C_{30}$ [15,16]. In addition, the authors [17] published SLR-based $C_{20}$ solutions specifically for GRACE RL06 data published by Deutsches GeoForschungsZentrum (GFZ), German Research Centre for Geosciences. Apparently, different low degree zonal harmonic coefficients may cause discrepancies among GRACE-derived mass redistribution. It was recognized that these low degree zonal harmonics derived from SLR have non-negligible impacts on accurately deriving mass redistribution from GRACE data [15,18].

Accurately deriving mass redistribution from GRACE data also necessitates the inclusion of geocenter motion, which depends only on the degree one coefficients of surface load [19–22]. As GRACE alone cannot provide the degree one coefficient changes, the authors of [23] computed degree one coefficients through the combination of GRACE and outputs from the ocean models in order to improve the accuracy of mass variations derived from GRACE data. Based on their method, monthly estimates of degree one coefficients (hereafter, we call them SW) were prepared for GRACE RL05 data. The authors of [24] improved degree one coefficient estimates by optimizing the processing choices (i.e., the truncation of spherical harmonic coefficients, the width of coastal buffer zone), and for the first time by including a gravitational self-consistent mass redistribution. Estimates of degree one coefficients following [24] are now provided in TN13 for GRACE RL06 data published by each data processing center. As the optimized method utilized GRACE spherical harmonic coefficients of degree two and higher and GRACE-determined $C_{20}$ were replaced with SLR-based $C_{20}$ prior to the estimation of the degree one coefficients, there were two versions of degree one coefficients (hereafter, we call them SUN11, SUN14) separately corresponding to SLR-based $C_{20}$ estimates in TN11 and TN14 for GRACE RL06 data published by each data processing center. Besides, [25] also reported improved estimates of degree one coefficients (referred to as SV) by combining time-variable gravity and ocean model outputs. Currently, it is unknown whether these improved estimates of degree one coefficients would cause discrepancies in GRACE-derived mass redistribution. The differences among mass change caused by the improved degree one coefficients and those previously derived from [23] are also unknown. Consequently, it is worthwhile to study the potential influence of these estimates of degree one terms on GRACE-derived mass redistribution.

In this study, we first compute mass change separately caused by degree one terms, SLR-based $C_{20}$, and GRACE-determined $C_{30}$ estimates during the GRACE mission span, as well as GSFC $C_{30}$ since March 2012. We then quantify the impacts of these low degree spherical harmonic coefficients corresponding to GRACE RL05 and RL06 data on mass change over the Greenland ice sheet (GrIS) and the Antarctic ice sheet (AIS), respectively. We finally assess the total influence of these low degree

spherical harmonic coefficients on polar ice sheet mass variations. Our study could provide a reference for quantifying the impacts of these low degree spherical harmonic coefficients on GRACE-derived polar ice sheet mass variations.

## 2. Data and Methods

GRACE L2 RL05 and RL06 monthly gravity field solutions were published by the Center for Space Research (CSR) at the University of Texas at Austin [26,27], GFZ [28,29] and the Jet Propulsion Laboratory (JPL) at the California Institute of Technology [30,31]. All the solutions are provided as fully normalized spherical harmonics with degree and order up to at least 60. In order to ensure the proper scientific applications of GRACE monthly gravity field solutions, some appropriate corrections associated with degree one terms, $C_{20}$, and glacial isostatic adjustment (GIA) are routinely applied. For GRACE RL06 data, some recent studies adopted the GSFC $C_{30}$ product in TN14 instead of GRACE-determined $C_{30}$ [15,16].

In this study, we only calculate mass variations caused by degree one terms (for each data processing center), SLR-based $C_{20}$ estimates, and GSFC $C_{30}$, as well as GRACE-determined $C_{30}$ estimates. As provided in Table 1, to assess the influence of degree one terms on polar ice sheet mass variations, monthly estimates of degree one terms based on the optimized method by [24] in two versions, namely SUN11 and SUN14, and those improved estimates (SV) published by [25], are analyzed for GRACE RL06 data from each data processing center. To compare the impact of degree one terms on polar ice sheet mass variations derived from GRACE RL05 data with that obtained from GRACE RL06 data, monthly estimates of degree one terms (SW) determined by [23] at the University of South Florida are also adopted. SLR-based $C_{20}$ solutions corresponding to GRACE L2 RL05 and RL06 data are separately analyzed during the GRACE mission span. For GRACE RL05 data, SLR-derived $C_{20}$ estimates were available in GRACE TN07 [11]. For GRACE RL06 data, SLR-based $C_{20}$ estimates in two versions, namely $C_{20}$ estimates derived from SLR only and those obtained from the combination of GRACE and SLR, are separately provided in GRACE TN11 [7] and TN14 [13]. Besides, SLR-based $C_{20}$ solutions, newly published by [17] (referred to as GFZ-SLR) at GFZ, are also utilized. In addition, the potential influence from $C_{30}$ is also studied, and GRACE-determined $C_{30}$ in RL05 and RL06 data from each data processing center and GSFC $C_{30}$ in GRACE TN14 since March 2012 are separately analyzed.

**Table 1.** The data sets adopted in this study. Only auxiliary data including degree one terms, satellite laser ranging (SLR)-based $C_{20}$ and Gravity Recovery and Climate Experiment (GRACE)-determined $C_{30}$ estimates separately corresponding to GRACE RL05 and RL06 data (from each data processing center) are analyzed during the GRACE mission span. Besides, SLR-based (GSFC) $C_{30}$ estimates since March 2012 are also utilized. Note that the following abbreviations are used: TN—GRACE Technical Note; SW—degree one terms from [23]; SUN11—degree one terms from [24], with $C_{20}$ estimates in TN11 used; SUN14—degree one terms from [24], with $C_{20}$ estimates in TN14 used; SV—degree one terms from [25], with $C_{20}$ estimates in TN14 used; GFZ-SLR—$C_{20}$ estimates from [17]; GSFC—the Goddard Space Flight Center.

| Data Used | Degree One Terms | $C_{20}$ | $C_{30}$ |
|---|---|---|---|
| GRACE RL05 (CSR/GFZ/JPL) | SW | TN07 | GRACE |
| GRACE RL06 (CSR/GFZ/JPL) | SUN11, SUN14, SV | TN11, TN14, GFZ-SLR | GRACE, GSFC |

We conduct four experiments to derive mass variations caused by these low degree spherical harmonic coefficients separately over the GrIS and the AIS: 1) only degree one terms corresponding to GRACE RL05 and RL06 data are utilized to compute mass variations; 2) only SLR-based $C_{20}$ estimates from TN07, TN11, TN14, and GFZ are separately used to calculate mass variations; 3) only GRACE-determined $C_{30}$ in RL05 and RL06 data and SLR-based $C_{30}$ in TN14 are used to compute

mass variations; 4) mass variations caused by degree one terms and the corresponding SLR-based $C_{20}$ estimates together with GRACE-determined $C_{30}$ or GSFC $C_{30}$ are also computed for RL05 and RL06 data. For experiment 1, SW only stands for one time series of degree one terms, as the correction from degree one terms is the same for all the GRACE RL05 data published by each data processing center. SUN11 (or SUN14 or SV) actually indicates three time series of degree one terms separately corresponding to GRACE RL06 data provided by CSR, GFZ, and JPL, since the correction from degree one terms is not exactly the same for all the GRACE RL06 data published by each data processing center. We calculate the correction from degree one terms for GRACE RL06 data published by each data processing center. In experiment 2, there is only one time series of SLR-based $C_{20}$ estimates in TN07 (or TN11, or TN14, or GFZ-SLR) for all the GRACE data published by each data processing center. Only one time series of GSFC $C_{30}$ estimates since March 2012 are provided in TN14 and then adopted in experiment 3. GRACE-determined $C_{30}$ solutions separately corresponding to GRACE RL05 and RL06 data published by each data processing center are analyzed in experiment 3. As for the analysis of the combination of degree one terms, SLR-based $C_{20}$ estimates and GSFC or GRACE-determined $C_{30}$, three cases are considered: 1) for GRACE RL05 data, SW, SLR-based $C_{20}$ estimates in TN07 and GRACE-determined $C_{30}$ solutions from each data processing center are used. 2) For GRACE RL06 data, SUN11, SLR-based $C_{20}$ estimates in TN11 and GRACE-determined $C_{30}$ solutions from each data processing center are analyzed. 3) Again, for GRACE RL06 data, SUN14, SLR-based $C_{20}$ estimates in TN14 and GRACE $C_{30}$ during the time span from April 2002 to February 2012, as well as GSFC $C_{30}$ solutions since March 2012 are utilized.

It should be noted that only these low degree spherical harmonics are used to calculate mass variations over both ice sheets. There are no north–south strips on the maps of low degree term-derived mass change so no Gaussian smoothing is needed. No GIA model is needed in this study, as we do not derive total mass variations over both ice sheets, avoiding the potential contamination from uncertainties of the GIA models [32] and high degree spherical harmonic coefficients in GRACE monthly gravity field solutions [33].

The study regions we choose are marked in Figure 1. We select the whole GrIS, including its surrounding small glaciers and ice caps (GICs), mostly considering that it is unlikely to separate mass variations of the ice sheet from the GICs, based on GRACE data with a relatively coarse spatial resolution of ~3 km or longer. For Antarctica, the regions with grounded ice sheets are chosen in this study.

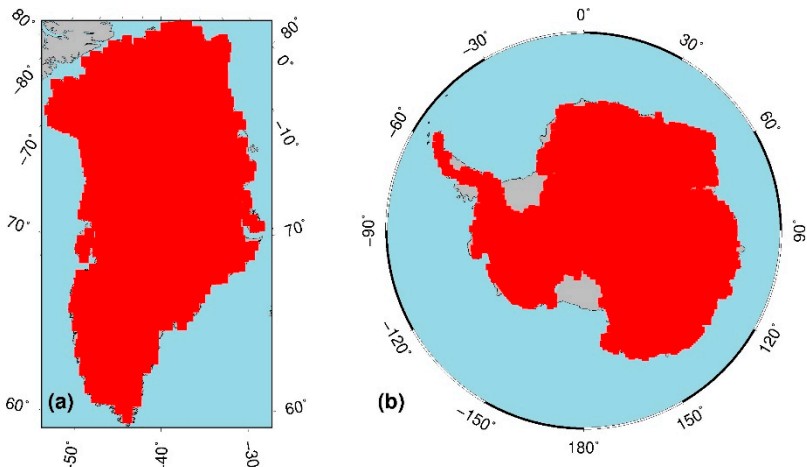

**Figure 1.** Geographic locations of the study regions in red: (**a**) the Greenland ice sheet (GrIS) and (**b**) the Antarctic ice sheet (AIS).

## 3. Results

We first compare the mass change time series computed from degree one terms separately corresponding to GRACE RL05 and RL06 data over each ice sheet (experiment 1), as depicted by Figure 2. It is obvious that these time series of mass change caused by degree one terms all exhibit significant annual variations together with a negative linear trend for the GrIS (Figure 2a) but a positive trend for the AIS (Figure 2b). Over the GrIS, annual peaks occurred in March/April/May and minima typically appeared in August/September/October. In contrast, annual peaks appeared in August/September/October over the AIS, with minima occurring in March/April/May. Apparent discrepancies can be seen among mass change time series derived from degree one terms corresponding to GRACE RL05 and RL06 data. For instance, mass change time series caused by degree one terms (SW) corresponding to GRACE RL05 data (blue lines in Figure 2) show smaller annual amplitudes and a more moderate linear trend than those for GRACE RL06 data (red, green, and black lines) over each ice sheet. All the mass change time series derived from SUN11, SUN14, and SV are in better agreement with each other. As listed in Table 2, mass variations caused by degree one terms (SW) corresponding to GRACE RL05 data show a linear trend of $-2.6 \pm 0.2$ Gt/yr and $13.1 \pm 0.8$ Gt/yr over the GrIS and the AIS, respectively, with respective annual amplitudes of 25.8 Gt and 122.1 Gt. For GRACE RL06 data, the trends of mass variations caused by degree one terms (SUN11) over the GrIS and the AIS are from $-6.7 \pm 0.2$ Gt/yr to $-7.1 \pm 0.2$ Gt/yr and from $35.8 \pm 1.0$ Gt/yr to $37.5 \pm 0.9$ Gt/yr, respectively, which in absolute value is 2.5 times larger than that for GRACE RL05 data. The annual amplitude of mass change caused by degree one terms corresponding to GRACE RL06 data is more than 30% larger than that over each ice sheet for GRACE RL05 data. Mass variations caused by degree one terms (SUN14) and those (SV) provided by [25] agree well with those derived from degree one terms (SUN11) over each ice sheet (Table 2 and Figure 2). Degree one terms (SUN14) modify the trend of mass variations by $-0.4 \pm 0.3$ Gt/yr and $2.4 \pm 1.4$ Gt/yr over the GrIS and the AIS, respectively, as compared to SUN11. Mass variations computed from SV show the most negative and the most positive mass trends together with a relatively smaller annual amplitude over the GrIS and the AIS, respectively, as compared to those derived from SUN11 and SUN14.

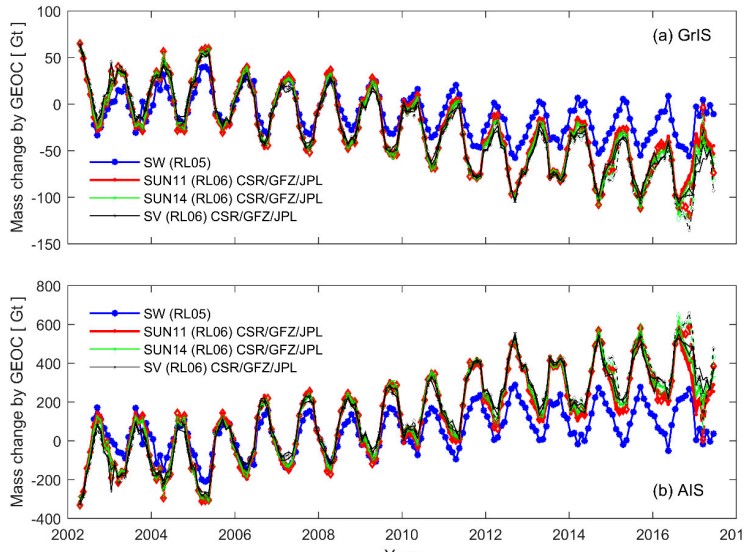

**Figure 2.** Time series of mass change caused by degree one terms (geocenter motion, GEOC) over (**a**) the GrIS and (**b**) the AIS, respectively, using degree one terms corresponding to GRACE RL05 and RL06 data published by each data processing center. CSR/GFZ/JPL indicates that the time series of mass change caused by degree one terms are calculated separately and are shown for GRACE data published by the Center for Space Research (CSR), Deutsches Geo Forschungs Zentrum (GFZ), and Jet Propulsion Laboratory (JPL).

**Table 2.** The linear trend and annual amplitude (Amp.) of mass change time series caused by GEOC or degree one terms over the GrIS and the AIS during the time span from August 2002 to July 2016, using degree one terms corresponding to GRACE RL05 and RL06 data published by CSR, GFZ, and JPL.

| 2002.08–2016.07 | | Mass Change Caused by GEOC | | | | | | | |
| | | RL05 | | RL06 | | | | | |
| | | SW | | SUN11 | | SUN14 | | SV | |
| | | Trend [Gt/yr] | Annual Amp. [Gt/yr] | Trend [Gt/yr] | Annual Amp. [Gt/yr] | Trend [Gt/yr] | Annual Amp. [Gt/yr] | Trend [Gt/yr] | Annual Amp. [Gt/yr] |
|---|---|---|---|---|---|---|---|---|---|
| GrIS | CSR | −2.6 ± 0.2 | 25.8 | −6.7 ± 0.2 | 36.4 | −7.0 ± 0.2 | 34.5 | −7.3 ± 0.2 | 30.2 |
| | GFZ | | | −7.1 ± 0.2 | 37.2 | −7.5 ± 0.2 | 35.4 | −8.1 ± 0.2 | 30.9 |
| | JPL | | | −6.8 ± 0.2 | 36.4 | −7.2 ± 0.2 | 34.5 | −7.5 ± 0.2 | 29.8 |
| AIS | CSR | 13.1 ± 0.8 | 122.1 | 35.8 ± 1.0 | 180.8 | 38.2 ± 1.0 | 169.0 | 39.4 ± 1.0 | 148.3 |
| | GFZ | | | 37.5 ± 0.9 | 185.1 | 40.0 ± 0.9 | 173.4 | 42.6 ± 0.9 | 151.3 |
| | JPL | | | 36.3 ± 1.0 | 180.8 | 38.7 ± 1.0 | 169.0 | 40.6 ± 1.0 | 146.1 |

To understand the impact of SLR-based $C_{20}$ estimates on mass variations of polar ice sheets, we compute mass variations over the GrIS and the AIS by only using SLR-based $C_{20}$ estimates during the GRACE mission span (experiment 2). As shown in Figure 3, mass variations caused by SLR-based $C_{20}$ estimates corresponding to GRACE RL05 and RL06 data are generally consistent with each other. Significant annual variations with a similar phase can be visible from all these mass change time series. Mass variations caused by $C_{20}$ estimates from TN07, TN11, and GFZ exhibit better agreement on the annual amplitude. A smaller annual amplitude can be seen from mass variations derived from $C_{20}$ estimates in TN14. As shown in Table 3, the annual amplitude of mass variations derived from $C_{20}$ estimates in TN14 is 25% smaller than those corresponding to TN07, TN11, and GFZ. The linear trend of mass variations derived from $C_{20}$ estimates in TN14 is more negative, with its absolute value more than 50% larger than those from $C_{20}$ estimates provided by TN07, TN11, and GFZ. Compared with $C_{20}$ estimates from TN11, TN14-provided $C_{20}$ estimates modified the linear trend of mass variations over the GrIS and the AIS by −1.9 ± 0.1 and −11.4 ± 0.5 Gt/yr, respectively, during the period from August 2002 to July 2016 (Table 3). GFZ-published $C_{20}$ estimates and those in TN11 have very similar impacts on the linear trend of mass variations over each ice sheet, i.e., over the AIS, the difference between the two mass trends is 0.7 Gt/yr.

We then examine the influence of $C_{30}$ from GRACE RL05, RL06, and GSFC on mass variations over both ice sheets (experiment 3). As shown in Figure 4, the mass change time series caused by $C_{30}$ from GRACE RL05, RL06, and GSFC are generally consistent with each other over ice sheets during the common study period, except those from CSR-published $C_{30}$ in GRACE RL05 show some differences during the end of the GRACE mission. If selecting the study period from March 2012 to July 2016, the annual amplitudes of mass change time series corresponding to $C_{30}$ from GRACE RL05 and RL06 data are 35 and 228 Gt, respectively, for the GrIS and the AIS, as compared to 27 and 175 Gt for those from SLR-based $C_{30}$ terms in TN14 from GSFC. From Figure 4, it can be seen that the $C_{30}$-caused mass change time series show a negative linear trend over the GrIS before 2012 but a positive trend after 2012. Over the AIS, a positive linear trend can be observed before 2012 but a negative trend after 2012. As provided in Table 4, the mean value of linear trends of mass change time series caused by $C_{30}$ from GRACE RL06 is −1.5 ± 0.5 and 9.7 ± 3.1 Gt/yr, respectively, for the GrIS and the AIS during the period from August 2002 to February 2012. It correspondingly becomes 1.7 ± 2.0 and −10.6 ± 12.2 Gt/yr during the time span from March 2012 to July 2016. It is noteworthy that

the impact of $C_{30}$ replacement on ice sheet mass variations derived from each data processing center differs after March 2012. Those differences modify the mass trend of the AIS by 16.3, 9.6, and 5.6 Gt/yr, respectively, for GRACE RL06 data published by CSR, GFZ, and JPL. If selecting the period from August 2002 to July 2016, as provided in Table 5, the replacement of $C_{30}$ terms after March 2012 affects the trend of mass variations over the GrIS and the AIS, respectively, by 0.2 and 1.0 Gt/yr.

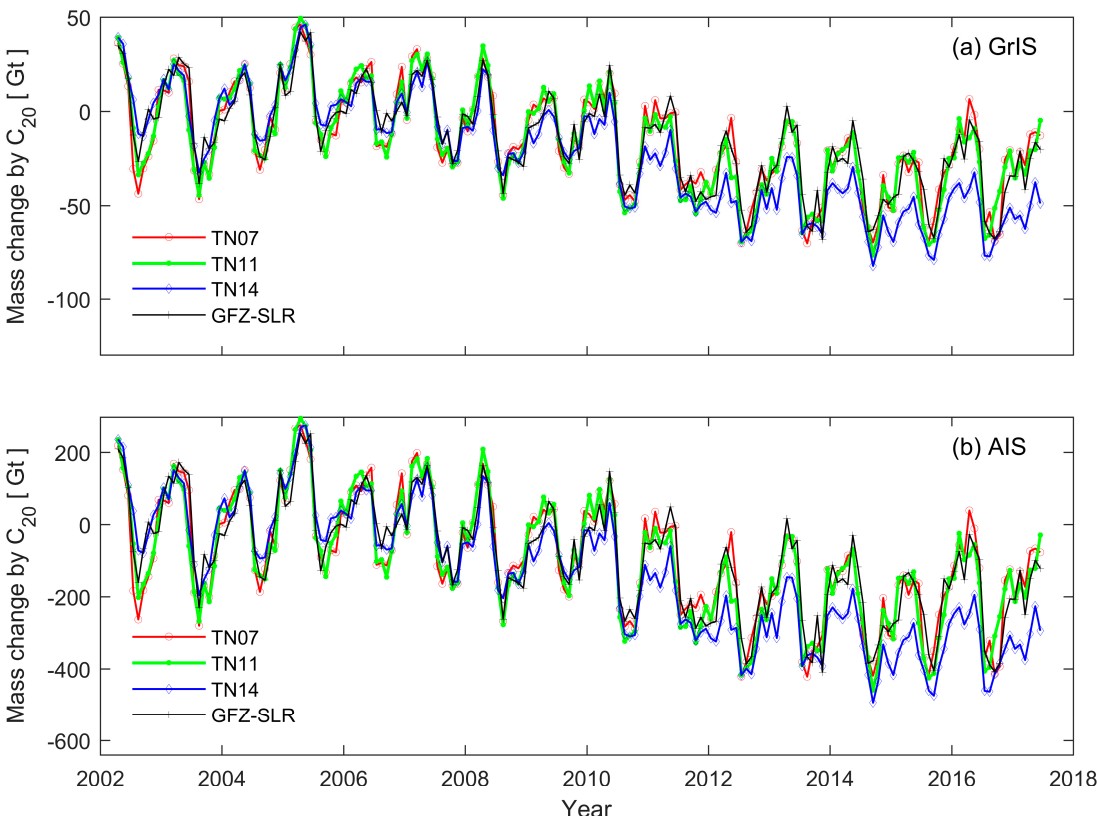

**Figure 3.** Mass change time series computed by only using SLR-based $C_{20}$ estimates from TN07, TN11, TN14, and GFZ over (**a**) the GrIS and (**b**) the AIS during the GRACE mission span.

**Table 3.** The linear trend, annual amplitude and annual phase of mass change time series caused by SLR-based $C_{20}$ estimates over the GrIS and the AIS during the time span from August 2002 to July 2016, using SLR-based $C_{20}$ estimates in TN07, TN11, TN14, and GFZ.

| 2002.08–2016.07 | | Mass Change Caused by SLR-based $C_{20}$ Estimates | | |
| --- | --- | --- | --- | --- |
| | | Trend [Gt/yr] | Annual Amp. [Gt] | Annual Phase [Degree] |
| GrIS | TN07 | −3.8 ± 0.2 | 24 | 8 |
| | TN11 | −4.1 ± 0.2 | 25 | 10 |
| | TN14 | −6.0 ± 0.2 | 16 | 3 |
| | GFZ-SLR | −4.0 ± 0.2 | 20 | 356 |
| AIS | TN07 | −22.6 ± 1.4 | 146 | 8 |
| | TN11 | −24.7 1.5 | 152 | 10 |
| | TN14 | −36.1 ± 1.3 | 97 | 3 |
| | GFZ-SLR | −24.0 ± 1.3 | 122 | 356 |

Finally, we compare mass change time series computed by the combination of degree one terms, the corresponding SLR-based $C_{20}$ and $C_{30}$ estimates over both ice sheets (experiment 4). As depicted by Figure 5, for combinations including the same SLR-based $C_{20}$ estimates, mass change time series from the three data processing centers agree well with each other, except for the time span after July

2016. The linear trend of mass variations computed from degree one terms, SLR-based $C_{20}$ and $C_{30}$ estimates in TN14 (green lines), is more negative over the GrIS, as compared to those associated with TN07 and TN11 corresponding to GRACE RL05 and RL06 data after March 2012. The mean value of linear trends of mass variations caused by these low degree spherical harmonics corresponding to GRACE RL05 (SW + TN07 + GRACE $C_{30}$), RL06 (SUN11 + TN11 + GRACE $C_{30}$), and RL06 (SUN14 + TN14 + GSFC) is $-8.1 \pm 0.7$, $-12.3 \pm 0.7$, $-14.7 \pm 0.7$ Gt/yr over the GrIS, respectively (Table 6). That is, the trend of mass variations computed from SUN11 + TN11 + GRACE $C_{30}$ in absolute value is 50% larger than SW + TN07 + GRACE $C_{30}$ over the GrIS. Compared with SUN11 + TN11 + GRACE $C_{30}$, the linear trend of mass variations caused by SUN14 + TN14 + GSFC in absolute value is 20% larger over the GrIS. Over the AIS, as provided by Table 6, the signs of linear trends of mass variations caused by these low degree terms corresponding to GRACE RL05 (SW + TN07 + GRACE $C_{30}$) for CSR, GFZ, and JPL are not the same, while more consistent linear trends can be seen from mass change time series corresponding to GRACE RL06 (SUN11 + TN11 + GRACE $C_{30}$), RL06 (SUN14 + TN14 + GSFC). The difference between the two positive linear trends of the mass change time series caused by these low degree terms corresponding to GRACE RL06 (SUN11 + TN11 + GRACE $C_{30}$) and RL06 (SUN14 + TN14 + GSFC) is about 8 Gt/yr over the AIS.

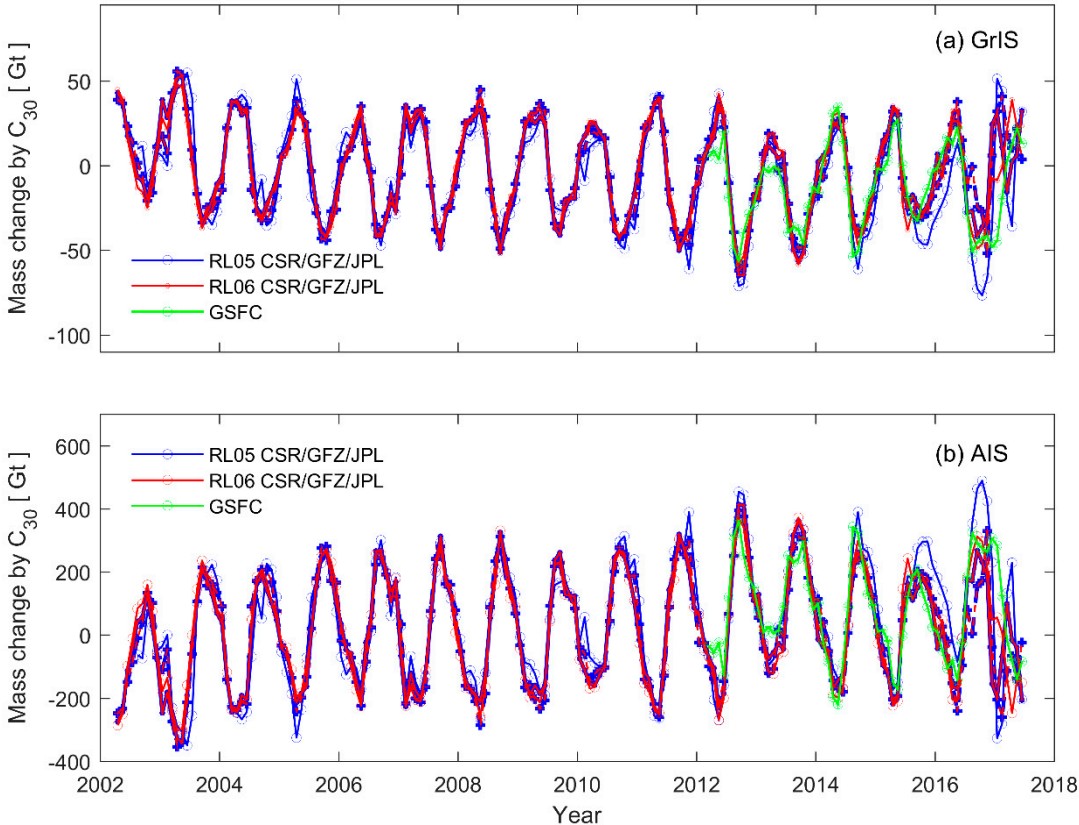

**Figure 4.** Mass change time series caused by $C_{30}$ from CSR/GFZ/JPL-published GRACE RL05 and RL06 data and by SLR-based $C_{30}$ in TN14 from GSFC over (**a**) the GrIS and (**b**) the AIS during the GRACE mission span. CSR/GFZ/JPL means that time series of mass change caused by $C_{30}$ are separately calculated and shown for GRACE data published by CSR, GFZ, and JPL.

**Table 4.** The trends of mass change time series caused by $C_{30}$ estimates separately from GRACE RL05/RL06 data and GSFC over the GrIS and the AIS during the period from August 2002 to February 2012, and the period from March 2012 to July 2016.

| Mass Change Caused by $C_{30}$ | | RL05 | | RL06 | | |
| --- | --- | --- | --- | --- | --- | --- |
| | | Trend [Gt/yr] | | Trend [Gt/yr] | | |
| | | GRACE | | GRACE | | GSFC |
| | | 2002.08–2012.02 | 2012.03–2016.07 | 2002.08–2012.02 | 2012.03–2016.07 | 2012.03–2016.07 |
| GrIS | CSR | −2.6 ± 0.4 | −2.2 ± 1.2 | −1.1 ± 0.3 | 0.8 ± 1.2 | 3.3 ± 1.0 |
| | GFZ | −1.4 ± 0.3 | 1.3 ± 1.0 | −1.5 ± 0.3 | 1.8 ± 1.0 | 3.3 ± 1.0 |
| | JPL | −1.9 ± 0.3 | 2.8 ± 1.1 | −2.0 ± 0.3 | 2.4 ± 1.2 | 3.3 ± 1.0 |
| | Mean | − | − | −1.5 ± 0.5 | 1.7 ± 2.0 | 3.3 ± 1.0 |
| AIS | CSR | 16.8 ± 2.5 | 13.9 ± 7.8 | 7.0 ± 1.7 | −4.8 ± 7.4 | −21.1 ± 6.2 |
| | GFZ | 9.2 ± 1.9 | −8.1 ± 6.6 | 9.6 ± 1.9 | −11.5 ± 6.3 | −21.1 ± 6.2 |
| | JPL | 11.8 ± 1.8 | −17.6 ± 7.2 | 12.5 ± 1.8 | −15.5 ± 7.4 | −21.1 ± 6.2 |
| | Mean | − | − | 9.7 ± 3.1 | −10.6 ± 12.2 | −21.1 ± 6.2 |

**Table 5.** The trend of mass variations caused by $C_{30}$ from GRACE RL05, RL06, and GSFC over the GrIS and the AIS during the period from August 2002 to July 2016.

| 2002.08–2016.07 | | Trends of Mass Change Caused by $C_{30}$ (RL05) [Gt/yr] | Trends of Mass Change Caused by $C_{30}$ (RL06) [Gt/yr] | |
| --- | --- | --- | --- | --- |
| | | GRACE | GRACE | GSFC |
| GrIS | CSR | −2.5 ± 0.2 | −1.2 ± 0.2 | −1.4 ± 0.2 |
| | GFZ | −1.4 ± 0.2 | −1.3 ± 0.2 | −1.5 ± 0.2 |
| | JPL | −1.5 ± 0.2 | −1.5 ± 0.2 | −1.6 ± 0.2 |
| AIS | CSR | 15.7 ± 1.4 | 7.9 ± 1.1 | 8.6 ± 1.2 |
| | GFZ | 8.9 ± 1.1 | 8.4 ± 1.1 | 9.5 ± 1.2 |
| | JPL | 9.4 ± 1.1 | 9.4 ± 1.2 | 10.2 ± 1.2 |

**Table 6.** The trends of mass change time series caused by the combination of degree one terms, SLR-based $C_{20}$ and $C_{30}$ estimates corresponding to GRACE RL05 and RL06 data over the GrIS and the AIS during the period from August 2002 to July 2016.

| 2002.08–2016.07 | | Trends of Mass Change Caused by Degree One Terms, $C_{20}$ and $C_{30}$ [Gt/yr] | | |
| --- | --- | --- | --- | --- |
| | | RL05 (SUN11 + TN11 + GRACE $C_{30}$) | RL06 (SUN11 + TN11 + GRACE $C_{30}$) | RL06 (SUN14 + TN14 + GSFC) |
| GrIS | CSR | −8.8 ± 0.4 | −12.0 ± 0.4 | −14.4 ± 0.4 |
| | GFZ | −7.6 ± 0.4 | −12.6 ± 0.4 | −15.0 ± 0.4 |
| | JPL | −7.8 ± 0.4 | −12.4 ± 0.4 | −14.8 ± 0.4 |
| | Mean | −8.1 ± 0.7 | −12.3 ± 0.7 | −14.7 ± 0.7 |
| AIS | CSR | 6.2 ± 2.0 | 19.0 ± 1.5 | 10.8 ± 1.5 |
| | GFZ | −1.1 ± 1.7 | 21.2 ± 1.6 | 13.4 ± 1.6 |
| | JPL | −0.1 ± 1.7 | 20.9 ± 1.7 | 12.8 ± 1.7 |
| | Mean | 1.7 ± 3.1 | 20.4 ± 2.8 | 12.3 ± 2.8 |

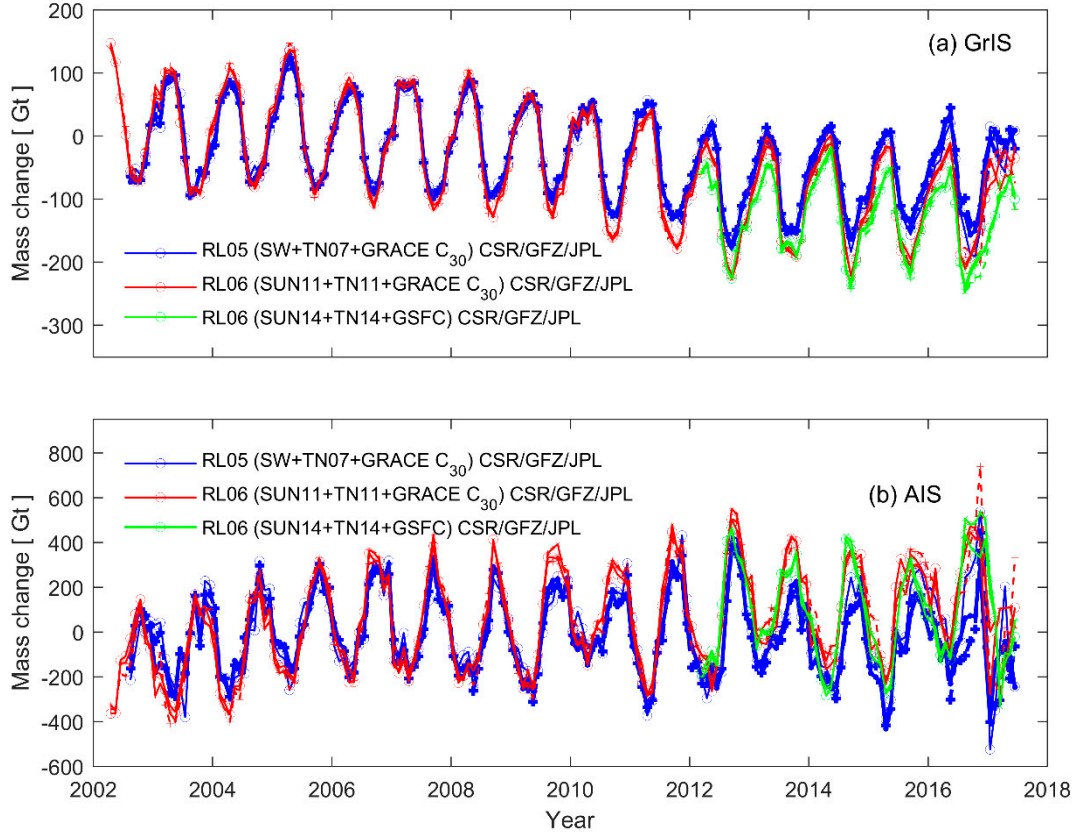

**Figure 5.** Mass change time series computed by the combination of degree one terms, SLR-based $C_{20}$ estimates, and $C_{30}$ solutions corresponding to CSR/GFZ/JPL-published GRACE RL05 and RL06 data over (**a**) the GrIS and (**b**) the AIS during the GRACE mission span. CSR/GFZ/JPL means time series of mass change caused by the combination of degree one terms, SLR-based $C_{20}$ and GRACE-determined $C_{30}$ estimates from CSR, GFZ, and JPL, are separately calculated and shown.

## 4. Discussion

The accuracy of GRACE-derived ice sheet mass balance estimates is affected by many factors, such as the post-processing techniques used to remove the north–south stripes in the spatial domain, the uncertainty of the adopted GIA model, the influences from degree one and SLR-based $C_{20}$ time series, and so on. Currently, the uncertainty of the GIA model has been recognized as the major source of uncertainty in ice mass balance. For instance, over the AIS, [32] pointed out that the uncertainties of six commonly used GIA models range from ±13 Gt/yr to ±27 Gt/yr. The contribution from low degree spherical harmonics was thought to have a non-negligible effect on GRACE-derived mass variations since the early stage of the GRACE mission. To date, the choice of these low degree spherical harmonics for accurately deriving ice sheet mass balance estimates is still an open issue: there are different approaches used to compute estimates of these low degree spherical harmonic coefficients, which could cause discrepancies among GRACE-derived ice mass balance estimates.

Here, we select the AIS as an example region to demonstrate the influence of these low degree spherical harmonics on GRACE-derived ice mass variations. Table 7 provides mass balance estimates for the AIS reported by several previous studies, with the used data, the study period and the adopted low degree terms shown. In order to analyze the impact of low degree terms on ice mass balance estimates, we calculate the trend of mass variations caused by the respective degree one terms (including SW, SUN11, SUN14, and SV), and caused separately by SLR-based $C_{20}$ during the corresponding study period over the AIS. Apparently, compared with the impacts of degree one terms corresponding to GRACE RL05 data, significantly increased influences can be found from these degree one terms corresponding to GRACE RL06 data on the AIS mass balance estimates. The increased influences are

primarily associated with the optimized processing choices in [24,25] when estimating degree one terms, e.g., the truncation degree of GRACE gravity field solutions, the width of the buffer zone, and including self-attraction and loading effects. For GRACE RL06 data, it is obvious that differences among these three time series of degree one terms affect the AIS mass balance estimates by 3~6 Gt/yr, depending on the study period. SV-caused mass variations over the AIS show the most positive trend, while the SUN11-derived mass trend is the least. Compared with SLR-based $C_{20}$ in TN11, mass trends derived by SLR-based $C_{20}$ in TN14 are 30%~50% more negative (by exceeding 11 Gt/yr) over the AIS. It should be noted that the differences between SLR-based $C_{20}$ estimates in TN11 and TN14 are mainly attributed to the different data reduction arc lengths and the application of the GRACE-derived forward model [13]. The differences between $C_{20}$ estimates in TN07 and TN11 may result from the applied background models corresponding to GRACE RL05 and RL06 data, respectively, e.g., the modeled ocean and atmosphere signal using the Level-1B Atmosphere Ocean De-aliasing product, the mean pole definition and pole tides, and so on [34]. More details about the approaches to solve for SLR-based $C_{20}$ estimates can be found in [7,9,13]. Considering the major source of uncertainty in ice mass balance, namely, the uncertainty of the AIS mass balance estimates caused by GIA correction, the influences from differences among degree one terms or between SLR-based $C_{20}$ on the AIS mass balance estimates should be carefully taken into account. We do recommend the officially published SLR-based $C_{20}$ estimates in TN14 and the corresponding degree one terms. We should also recognize that it would be good to keep $C_{20}$ estimates from SLR only updated, as independently solved $C_{20}$ estimates could be reasonable for cross-validation. As for $C_{30}$, Table 5 illustrates that GSFC $C_{30}$ replacement has very limited impact on GRACE-derived mass variations over the AIS, if selecting GRACE data during the whole mission span. However, according to Table 4, during the period from March 2012 to July 2016, GSFC $C_{30}$ can modify the mass trend of the AIS by 16.3, 9.6, and 5.6 Gt/yr, respectively, for GRACE RL06 data published by CSR, GFZ, and JPL. That is, the replacement of $C_{30}$ has some impact on GRACE-derived mass variations over the AIS, depending on the study period and the data provided by which data processing center. According to [14], GSFC $C_{30}$ replacement was mainly recommended for applications of GRACE RL06 solutions during single accelerometer mode (after October 2016).

**Table 7.** The trends of mass variations over the AIS in previous studies and the contributions from low degree terms in this study. Note, mass trends caused by low degree terms corresponding to TN11, TN14, and SV for GRACE RL06 data over the AIS in this study are marked separately in light blue, black, and red. For GRACE RL05, the impacts of low degree terms on the trend of the AIS mass variations in our study are shown in black, with the corresponding results in previous studies shown in orange.

| Previous Studies | Data and Time Period | Low Degree Terms Used | Mass Trend over the AIS [Gt/yr] | Mass Trend Caused by Low Degree Terms in This Study [Gt/yr] | |
|---|---|---|---|---|---|
| | | | | Degree One Terms | SLR-Based $C_{20}$ |
| [35] | GRACE RL05 2003–2011 | Three independent GEOC time series; $C_{20}$ in TN07 | −83 ± 36 | 10.5 ± 1.5 13.2 | −15.7 ± 2.5 |
| [36] | GRACE RL05 2003.02–2013.12 | Degree one terms from SW; $C_{20}$ in TN07 | −83 ~ −108 | 15.3 ± 1.2 >19 | −24.8 ± 2.1 |
| [13] | GRACE RL06 2008–2015 | $C_{20}$ in TN11 $C_{20}$ in TN14 | −133.1 ± 5.1 −148.4 ± 5.5 | 40.3 ± 3.6 42.9 ± 3.5 43.6 ± 3.9 | −34.3 ± 3.0 −45.6 ± 2.6 |
| [15] | GRACE RL06 and GRACE−FO (CSR) 2002.04–2019.09 | Degree 1 terms from SV $C_{20}$ in TN14 | −107 ± 55 | 34.0 ± 0.9 37.1 ± 0.9 40.3 ± 1.4 | −22.7 ± 1.3 −34.7 ± 1.1 |

## 5. Conclusions

In this study, we carry out four experiments to investigate mass variations caused by degree one terms, SLR-based $C_{20}$, and $C_{30}$ estimates corresponding, respectively, to GRACE RL05 and RL06

data published by CSR, GFZ, and JPL over the GrIS and the AIS. We report that the trend of mass change caused by degree one terms corresponding to GRACE RL06 data in absolute value is 2.5 times larger than that for GRACE RL05 data. The annual amplitude of mass change caused by degree one terms corresponding to GRACE RL06 data is more than 30% larger than that over each ice sheet for GRACE RL05 data. Mass variations caused by SLR-based $C_{20}$ estimates from TN07, TN11, and GFZ exhibit better agreement on the annual amplitude, with relatively smaller annual amplitude revealed by those corresponding to TN14. The linear trend of mass variations derived from $C_{20}$ estimates in TN14 is more negative, with its absolute value being more than 50% larger than those from $C_{20}$ estimates provided by TN07, TN11, and GFZ over each ice sheet. The mass change time series caused by SLR-based $C_{30}$ estimates in TN14 exhibit an annual amplitude which is 23% smaller than those derived from GRACE RL05 and RL06 data. The mean value of linear trend of mass change time series caused by SLR-based $C_{30}$ in TN14 in absolute value is almost two times larger than those from GRACE RL06 data during the time span from March 2012 to July 2016. The combination of degree one terms, SLR-based $C_{20}$, and $C_{30}$ estimates corresponding to TN14 modifies the trend of mass variations over the GrIS and the AIS by −2.4 and 8.0 Gt/yr, respectively, during the period from August 2002 to July 2016, compared to that corresponding to TN11. Our study demonstrates that these low degree terms have significantly increased impacts on mass variations over polar ice sheets. Compared with GRACE RL05 data, improvements can be seen in these consistent linear trends of mass variations computed by the combination of these low degree terms corresponding to GRACE RL06 over the AIS during the period from August 2002 to July 2016. This study suggests that reliable low degree spherical harmonic coefficients are crucial for the scientific applications of GRACE/GRACE-FO gravimetry data.

**Author Contributions:** X.S. conceptualized the experiments. She also performed the analysis and wrote the manuscripts. J.G. helped with improving the design of the experiments, and contributed to the analysis of the results. C.K.S., Z.L., and Y.Z. provided useful comments and helped polish the manuscript. All authors have read and agreed to the published version of the manuscript.

**Funding:** This research is funded by the National Key R&D Program of China, grant number 2018YFC1503503, and the National Natural Science Foundation of China, grant number 41804014, 41931074, and 41974040. It is also partially supported by National Key Research & Development Program of China (2017YFA0603103) and Strategic Priority Research Program of the Chinese Academy of Sciences (XDA19070302).

**Acknowledgments:** X.S. would like to thank the platform support from the National Precise Gravity Measurement Facility. X.S. and Z.L. give thanks for the administrative help from the Center for Gravitational Experiments, Huazhong University of Science and Technology. All the authors thank the three anonymous reviewers for their constructive comments which led to the improvement of the manuscript. In this study, GRACE-determined $C_{30}$ data products are from NASA via CSR, GFZ, and JPL, which can be accessed at http://podaac.jpl.nasa.gov/grace and http://isdc.gfz-potsdam.de/grace. The auxiliary data for the corrections of degree one terms, SLR-based $C_{20}$ and $C_{30}$ estimates are also available through the above two websites, e.g., http://isdc.gfz-potsdam.de/grace/DOCUMENTS/ TECHNICAL_NOTES/. Degree one terms provided by Sutterley and Velicogna (2019) can be downloaded from https://doi.org/10.6084/m9.figshare.7388540. GFZ-provided SLR-based $C_{20}$ solutions are available from http://isdc.gfz-potsdam.de/grace. Some figures in the paper are generated by using the Generic Mapping Tool (GMT) [37].

**Conflicts of Interest:** The authors declare no conflict of interest.

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
