# Peer review of "Increased Low Degree Spherical Harmonic Influences on Polar Ice Sheet Mass Change Derived from GRACE Mission"

_remotesensing, doi:10.3390/rs12244178_

Round 1
Reviewer 1 Report
This manuscript is clear, well written, and the methods are sound. Useful work was performed but I dont think it rises to the level of a scientific paper.
Major: this manuscript does not provide especially useful new results. Yes, there is a difference between older and newer versions of the degree 1, 2 and 3 corrections to GRACE, and yes, they affect estimates of trends in Antarctica. This is to be expected. Quantifying them is somewhat useful, but in itself alone does not justify a scientific paper. There is no validation with independent data, there is no justification for which might be ‘better’.
In addition there are issues with the manuscript itself:
- TN11, TN14 are mentioned many times, central to the discussion, but never cited. This reviewer knows what they are but the average reader will NOT, you MUST give a reference, a web link to the documents.
- SUN11, SUN14 are used many times but never defined or referenced.
Reviewer 2 Report
Based on data from the GRACE mission, the relationship between low-degree spherical harmonics and changes in the polar ice sheet mass is analyzed. Variations in the mass of the Greenland ice sheet and the Antarctic ice sheet were estimated using different data sets. Quantitative estimates of the impact of ice sheet mass variation on low-degree spherical harmonic coefficients obtained from GRACE data are determined.
The paper compares the coefficients C20 and C30 obtained from SLR data and data from the GRACE mission. A reasonable conclusion is made that it is more reliable to use low harmonics based on satellite gravimetry, rather than laser location, to estimate the mass of the ice cover.
The result of the study is the assumption of the authors that spherical harmonic coefficients of low degree give more reliable values for solving the problem. This information is crucial when using the GRACE/GRACE-FO mission data.
We believe that the reviewed article can be recommended for publication.
Reviewer 3 Report
Review of the paper „Increased Low Degree Spherical Harmonic Influences on Polar Ice Sheet Mass Change Derived from GRACE Mission” submitted to Remote Sensing.
The paper assesses the impact of replacing spherical harmonics of the time-variable gravity field C20 and C30 in GRACE-based solutions by external sources based, e.g., on satellite laser ranging (SLR) data. The impact of replacement C30 is different in this paper than it was shown in previous studies, e.g., by Loomis et al (2019 https://doi.org/10.1029/2019GL082929). However, different periods were used in these two studies. The paper is well written, however, some issues have to be resolved before the publication.
Major issues
- Figure 3 shows that the series TN14 is clearly different than other series in terms of the secular trend. This issue needs a detailed discussion of the possible origin of the different rates. In the SLR series, the same background models as in the GRACE solution have to be applied. These include AOD, mean pole definition, and tidal displacements (pole tides, ocean pole tide, etc.). Any inconsistencies in the background model and truncation of the SLR solutions up to a certain degree or order may change the signal in the derived series, see (Cheng and Ries, 2017: https://doi.org/10.1007/s00190-016-0995-5)
- The replacement of SLR-derived C20 or C30 and keeping all other gravity field coefficients as they are derived from GRACE is not a proper solution. This approach neglects the correlations between different spherical harmonics in both SLR and GRACE solutions. A much better approach would be the combination of the normal equations that consider the full variance-covariance information from both solutions. This approach was proposed by Sosnica et al (2015: https://doi.org/10.1007/s00190-015-0825-1), who also studied differences between the GRACE and the SLR-derived higher-order gravity field coefficients. Therefore, this approach should also be part of the discussion in the paper.
- C20 is strongly correlated with –C40, with C60, -C80, etc. Estimating C40 or not may change the signal in C20. Thus, replacing only one spherical harmonic coefficient from a solution with a different truncation may lead to biased estimates of C20. Moreover, -C30 is also correlated with the Z component of the geocenter motion (C10) and with C50. Different handling of these coefficients or neglecting the time-variable nature may also bias the C30 values. Hence, replacing just one value in the GRACE series with SLR values is not a good idea. This approach is popular in the GRACE community, however, may lead to wrong results.
- The literature review should be extended. The analysis of the low-degree spherical harmonics and their impact on Greenland and Antarctica mass balance in different configurations were studied, e.g., by Bonin et al 2018: (https://doi.org/10.5194/tc-12-71-2018).
- Cheng and Ries (2017) discussed the importance of including LARES as an additional satellite, starting in 2012, to recover better C30 values. Are the LARES data also employed in SLR-based series in this study? LARES can improve the C30 estimates, however, the quality of C30 will be different in the time series of observations, because the pre-2012 solutions will be based mostly on LAGEOS-1 and LAGEOS-2 (and other geodetic satellites, such as Starlette, Stella, and Ajisai). The summary of the SLR-based solution and geodetic satellites is discussed by Pearlman et al. (2019) in a review paper “Laser geodetic satellites: a high-accuracy scientific tool”.
- Another important issue is the geocenter motion (degree-1 spherical harmonics). The geocenter values must be consistent in all solutions, as it may change the general signal in Greenland and Antarctic. The SLR-derived geocenter motion series have typically larger amplitudes of the annual signal than the GRACE-based values (the GRACE-based values cannot be determined using the K-band observations, thus, must be derived from kinematic orbits or reconstructed based on higher-order and higher-degree spherical harmonic coefficients). Therefore, a discussion on the consistency between the degree-1 series used in this study should also be added to the paper.
Round 2
Reviewer 1 Report
After reading the revised manuscript, and the responses to the reviews, I believe the manuscript is ok for publication. I noticed than in the reviews the a few English language problems crept in (eg, Nevertheless, such concern was still exist [11]; SLR only in like those in TN11 updated, etc). PLease fix, it should be easy.
Author Response
Response
Thanks for your comment. We carefully read the manuscript again and make some
modifications. Please find them in lines 15, 23, 42, 44-45, 52-54, 337.
Reviewer 3 Report
The authors addressed most of the issues from the original review report.
I have just one comment. The low-degree gravity field coefficients are based on SLR observations provided by the ILRS stations. The collected data is distributed for free to all users. Therefore, the official ILRS paper describing the SLR satellites should be acknowledged:
Pearlman, M., Arnold, D., Davis, M. et al. Laser geodetic satellites: a high-accuracy scientific tool. J Geod 93, 2181–2194 (2019). https://doi.org/10.1007/s00190-019-01228-y
Author Response
Response
Thank you for the comments. Yes, we totally agree that the official ILRS paper did provide a good description about the development and application of geodetic satellites. It should be acknowledged. Previously, we did not quote it, mostly considering that it was a descriptive study and no data analysis was performed. Now we add it, as readers may also want to know more about SLR technology. Modifications can be found in lines 44-45, 419-426. Thanks again for your time and comments.